# *Klebsiella pneumoniae* Lipopolysaccharides Serotype O2afg Induce Poor Inflammatory Immune Responses Ex Vivo

**DOI:** 10.3390/microorganisms9061317

**Published:** 2021-06-17

**Authors:** Matteo Bulati, Rosalia Busà, Claudia Carcione, Gioacchin Iannolo, Giuseppina Di Mento, Nicola Cuscino, Roberto Di Gesù, Antonio Palumbo Piccionello, Silvestre Buscemi, Anna Paola Carreca, Floriana Barbera, Francesco Monaco, Francesca Cardinale, Pier Giulio Conaldi, Bruno Douradinha

**Affiliations:** 1Istituto di Ricovero e Cura a Carattere Scientifico-Istituto Mediterraneo per i Trapianti e Terapie ad Alta, Specializzazione (IRCCS-ISMETT), 90127 Palermo, Italy; mbulati@ismett.edu (M.B.); rbusa@ismett.edu (R.B.); giannolo@ismett.edu (G.I.); gdimento@ismett.edu (G.D.M.); ncuscino@ismett.edu (N.C.); fbarbera@ismett.edu (F.B.); fmonaco@ismett.edu (F.M.); fcardinale@ismett.edu (F.C.); pgconaldi@ismett.edu (P.G.C.); 2Fondazione Ri.MED, 90133 Palermo, Italy; ccarcione@fondazionerimed.com (C.C.); rdigesu@fondazionerimed.com (R.D.G.); apcarreca@fondazionerimed.com (A.P.C.); 3Department of Biological, Chemical and Pharmaceutical Sciences and Technologies-STEBICEF, University of Palermo, 90133 Palermo, Italy; antonio.palumbopiccionello@unipa.it (A.P.P.); silvestre.buscemi@unipa.it (S.B.)

**Keywords:** *Klebsiella pneumoniae*, nosocomial infection, lipopolysaccharides, immune evasion, antimicrobial resistance, NF-κB

## Abstract

Currently, *Klebsiella pneumoniae* is a pathogen of clinical relevance due to its plastic ability of acquiring resistance genes to multiple antibiotics. During *K. pneumoniae* infections, lipopolysaccharides (LPS) play an ambiguous role as they both activate immune responses but can also play a role in immune evasion. The LPS O2a and LPS O2afg serotypes are prevalent in most multidrug resistant *K. pneumoniae* strains. Thus, we sought to understand if those two particular LPS serotypes were involved in a mechanism of immune evasion. We have extracted LPS (serotypes O1, O2a and O2afg) from *K. pneumoniae* strains and, using human monocytes ex vivo, we assessed the ability of those LPS antigens to induce the production of pro-inflammatory cytokines and chemokines. We observed that, when human monocytes are incubated with LPS serotypes O1, O2a or O2afg strains, O2afg and, to a lesser extent, O2a but not O1 failed to elicit the production of pro-inflammatory cytokines and chemokines, which suggests a role in immune evasion. Our preliminary data also shows that nuclear translocation of NF-κB, a process which regulates an immune response against infections, occurs in monocytes incubated with LPS O1 and, to a smaller extent, with LPS O2a, but not with the LPS serotype O2afg. Our results indicate that multidrug resistant *K. pneumoniae* expressing LPS O2afg serotypes avoid an initial inflammatory immune response and, consequently, are able to systematically spread inside the host unharmed, which results in the several pathologies associated with this bacterium.

## 1. Introduction

Antimicrobial resistance is considered the next silent pandemic. Recently, the World Health Organization has established that novel strategies against multidrug resistant pathogens are urgently needed, including the need for the Gram negative *Klebsiella pneumoniae* bacterium [1]. Following successful colonization and systemic spreading, *K. pneumoniae* can induce a wide range of pathologies such as pneumonia, liver abscesses and urinary infections [2]. This pathogen has shown a remarkable ability to develop resistance to several antibiotics and is currently responsible for several nosocomial infection cases, related mortality and morbidity and associated economic burden [2,3,4]. *K. pneumoniae* resistance to last resort antibiotic class carbapenems has been widely reported. In 2014, the carbapenem-resistant (CRE) *K. pneumoniae* were responsible for 351,000 bloodstream infections and 2,062,000 serious infections worldwide [5], with an estimated rate of mortality of 13.3% [6].

Lipopolysaccharides (LPS) are components of the bacterial outer membrane composed by an O-antigen, a core oligosaccharide and lipid A. Many pathogens use their capsular polysaccharides and structurally-modified LPS to escape host microbicidal mechanisms such as phagocytic and macrophage action [7]. As an example, *K. pneumoniae* can survive within macrophages due to its capsular polysaccharides [8], while downregulation of the latter increases bacterial phagocytosis by macrophages [9] and some host factors also downplay LPS protective action, such as CD36 [10]. During *K. pneumoniae* infections, LPS O-antigens can bind the complement component C3b and impairs complement-mediate killing and promotes bacterial survival [2]. However, LPS can induce inflammation through the binding of lipid A to TLR4 and the consequent initiation of an inflammatory cascade of the production of chemokines and cytokines to fight bacterial infection is induced [2,11,12]. Interestingly, different *K. pneumoniae* strains can induce immune response with diverse outcomes. A previous work using CRE *K. pneumoniae* strains A28006 and A54970 belonging, respectively, to sequence type (ST) 11 and ST437 showed that the A28006 strain elicited production of high levels of IL-1β, IL-12 and TNF-α and of pyroptotic cell death in murine macrophages, while the A54970 strain promoted the release of high levels of IL-10 and low levels of IL-1β production that were also a result of mouse macrophages [13]. Furthermore, macrophages which had contact with the A54970 strain and were later treated with *K. pneumoniae* LPS were still unable to produce IL-1β. Moreover, strain A54970 was also able to inhibit inflammasome activation due to its ability to promote IL-10 secretion; this results in bacterial survival and dissemination [13]. The different inflammatory phenotypes induced by these two CRE *K. pneumoniae* strains might be caused by either diverse capsular polysaccharides or lipopolysaccharides serotypes. Different O-antigen serotypes are defined by their molecular structure and currently there are 10 serotypes: O1/O8, O2a, O2ac, O2ae/O2aeh/O9, O2afg, O3, O4, O5, O7 and O12 [14]. Although the O1-antigen is more predominant in *K. pneumoniae* clinical isolates [15,16], it was observed that the O2-antigen is prevalent in multidrug resistant strains, in particular, its subtype O2afg, which confers improved survival in human serum [16,17]. Thus, we sought to understand if *K. pneumoniae* LPS O2afg-antigens are involved in a mechanism of immune evasion that helps CRE strains in establishing and proliferating themselves within the host.

## 2. Materials and Methods

### 2.1. K. pneumoniae Strains and Sequencing and General Reagents

In this work, we used the *K. pneumoniae* reference strains B5055 (O1), C5046 (O2a) and 6613 (O2afg) [14,18] (Statens Serum Institut, Copenhagen, Denmark). *K. pneumoniae* clinical strains were isolated at the Istituto Mediterraneo per i Trapianti e Terapie ad Alta Specializzazione (IRCCS-ISMETT) from several sources (blood, urinary infections, wound, rectal swab, etc.) during the period between 2011 and 2017 and identified using Vitek MS (Biomerieux, Bagno a Ripoli, Italy). Internal medical evaluation classified the cases as either colonization, i.e., presence of bacteria in patients without causing related pathologies, or infection, i.e., pathological markers or symptoms that are observed in patients. Their genomes were analyzed through next generation sequencing (NGS) [19,20,21,22,23]. Briefly, total bacterial DNA was purified using the QIAamp DNA Mini Kit (Qiagen, Hilden, Germany) and 1 ng of DNA was processed with the Nextera XT library (Illumina, San Diego, CA, USA). The resulting libraries were run in a sequencer Illumina MiSeq (Illumina) as 151 bp paired-end reads, which were later assembled with SPAdes Genome Assembler (version 3.6.1; CAB, Saint Petersburg University, Saint Petersburg, Russia). The sequences of the *K. pneumoniae* strains were uploaded in the NCBI database (BioProject ID PRJNA687927).

Unless diversely indicated, all reagents we used in this work were purchased from Sigma-Aldrich (Darmstadt, Germany).

### 2.2. Sequence Type and O-Antigen Serotype Determination

The STs and capsular loci (KL) were determined using the Institut Pasteur *Klebsiella* multilocus sequence typing (MLST) website (https://bigsdb.pasteur.fr/cgi-bin/bigsdb/bigsdb.pl?db=pubmlst_klebsiella_seqdef, accessed on 1 November 2020) and the O-antigens were identified using the software Kaptive (https://kaptive-web.erc.monash.edu/, accessed on 1 November 2020). The *K. pneumoniae* B5055 sequence was already available online (accession number SAMN01999147) and the remaining reference strain sequences were uploaded online together with those from the clinical isolates. In particular, the sequence of the protein aspartyl beta-hydroxylase (lipid A hydroxylase) LpxO (accession number WP_002914281.1) was matched against the sequence of the analogous protein of the different reference *K. pneumoniae* strains used in this work and was 100% identical for all sequences analyzed.

### 2.3. LPS ELISA and Immunoblot

Sera derived from IRCCS-ISMETT patients who suffered either a *K. pneumoniae* infection or colonization and negative controls were tested for immunogenicity against *K. pneumoniae* B5050 protein extract by enzyme-linked immunosorbent assay (ELISA), as detailed elsewhere [16,24]. We have identified sera with high levels of antibodies against *K. pneumoniae*, namely 42 infection cases and 22 colonization cases. Conversely, 10 sera samples from patients with no identified *K. pneumoniae* infection or colonization showed no antibodies against this pathogen. To assess the immunogenicity against LPS O1, O2a and O2afg antigens, we extracted the LPS from the *K. pneumoniae* reference strains B5055 (O1), C5046 (O2a), 6613 (O2afg) and from an ST258 clinical isolate (O2afg) using the LPS Extraction Kit (Intron, Seoul, South Korea) according to the manufacturer’s instructions. The LPS was quantified by silver staining as described elsewhere [25] by using a Pierce™ Silver Stain Kit (Thermo Fisher Scientific, Monza, Italy) in accordance with the provider’s recommendations. *K. pneumoniae* commercial O1-antigen was used as a standard (10 µg/lane as standard). The following concentrations were obtained: LPS B5055 O1, 341 ng/µL; LPS C5046 O2a, 379 ng/µL; LPS 6613 O2afg, 505 ng/µL; LPS ST258 O2afg, 477 ng/µL. The purity of the LPS was assessed via High-Pressure Liquid Chromatography (HPLC), as previously described [26] and the their values for LPS O1, O2a, O2afg (6613) and O2afg (ST258) were 92.26%, 93.70%, 94.71%, and 96.97%, respectively (Appendix A). Commercial *K. pneumoniae* LPS O1 was used as reference (Appendix A). Immunogenicity of different LPS was assessed by ELISA as previously described [16,24] and the coating of ELISA plates was performed with 0.5 µg/well of the purified LPS and using serial dilutions of sera of infection and colonization patients. Optical density was read at 450 nm. Since antigen-binding efficiency of ELISA plate surfaces may differ for different glycan moieties, which renders it particularly difficult to compare such diverse glycans, we performed a dot blot. Briefly, an Immuno-Blot^®^ polyvinylidene difluoride (PVDF) membrane (Bio-Rad, Hercules, CA, USA) was activated with methanol and permitted to dry. Different quantities of the used LPS were applied in the membrane (1, 0.5, 0.25 and 0.125 µg), which was blocked with a solution of bovine serum albumin (BSA) 3% in tween phosphate-buffered saline (TPBS) for 1 h. Sera from an infection case, a colonization case and a negative case were diluted 1:1000 in blocking solution and incubated overnight at 4 °C. The following day, the membranes were washed three times with TPBS for 10 min each and incubated with a peroxidase-conjugated anti-human IgG F(ab’)2 specific (Jackson ImmunoResearch, Ely, UK) at a 1:10.000 dilution, for 90 min. Following three washes with TPBS, membranes were developed using ClarityTM Western ECL Substract (Bio-Rad) and images acquired using the Bio-Rad ChemiDocTM MP Imaging System.

### 2.4. Immunological Assays

Venous blood was collected in K3EDTA tubes (Greiner Bio-One GmbH, Kremsmünster, Austria) and diluted 1:4 in RPMI 1640 medium supplemented with 1% penicillin/streptomycin, 10mM HEPES (Euroclone, Pero, Italy) and 1mM L-glutamine (Lonza Group Ltd., Basel, Switzerland). Samples were stimulated for either 6 or 48 h with 1 µg/mL of *K. pneumoniae* LPS serotypes O1, O2a or O2afg (from both 6613 strain and ST258 clinical isolate). As control, commercial LPS from *Escherichia coli* 0111:B4 was used. For inhibition assays, blood samples were pre-incubated with 10 µg/mL of polymyxin B (PMB) for 30 min at 37 °C in a 5% CO_2_ atmosphere [27]. After incubation with LPS, levels of pro-inflammatory cytokines and chemokines present in the supernatants were assessed using magnetic beads technology from LuminexTM with the ProcartaPlex Multiplex Immunoassay (30 plex; Affymetrix, Vienna, Austria) [28]. Cellular viability was assessed by plating the cells in 96 well non-treated plates (Corning, New York, NY, USA) and, after the indicated time, the cells were collected and lysed with Cell Titer Glo Luminescent Cell Viability Assay reagent (Promega, Mannheim, Germany), in accordance with the provider’s indications. Luminescence was analyzed using the Spark Microplate Reader (Tecan, Männedorf, Switzerland) as described elsewhere [29].

Purification of monocytes was performed as previously described [30]. Briefly, peripheral blood mononuclear cells (PBMCs) were isolated from venous blood by density gradient centrifugation on Lympholyte Cell Separation Media (Cedarlane Laboratories Limited, Burlington, ON, Canada) and adjusted to 1 × 10^6^ cells/mL in the RPMI 1640 medium supplemented, as described above. CD14^+^ monocytes were separated from PBMCs by immunomagnetic sorting using anti-CD14 magnetic microbeads (MACS CD14 Microbeads, Miltenyi Biotec, Auburn, CA, USA). The CD14^+^ monocytes obtained from immunomagnetic sorting displayed a purity yield higher than 98%, which was determined by flow cytometry analysis. Stimulation of purified CD14^+^ monocytes with *K. pneumoniae* LPS, NF-κB inhibition with PMB and pro-inflammatory cytokines and chemokines quantification were performed as described above for whole blood assays.

In order to assess the levels of phagocytosis of *K. pneumoniae* bacteria expressing different O-antigens, reference strains B5055, C5046 and 6613 and the ST258 clinical isolate were grown overnight in MacConkey broth at 37 °C with agitation. During the following day, these strains were labelled with a pH-sensitive fluorescent probe that acts as a phagocytosis marker [31]. Labelling was done using the pHrodo™ Red Phagocytosis Particle Labeling Kit for Flow Cytometry (Thermo Fisher Scientific). To perform both labelling of the strains and the phagocytosis assay, the manufacturer’s instructions were precisely followed. Briefly, after the labelling of the bacteria, heparinized whole blood was collected and stored on ice up to 2 h prior to the phagocytosis assay. Twenty μL of pHrodo™ red stained bacteria were added to 100 μL of whole blood in sterile 5 ml conical tubes (1:10 pathogen to whole blood ratio). As a negative control, 20 μL of reconstitution buffer were added to 100 μL whole blood. The tubes were incubated at 37°C for 60 min and placed on ice to stop the reaction. One hundred μL of lysis buffer were added for 5 min at room tem-perature, followed by the addition of 1 ml of reconstitution buffer for 5 min at RT. Samples were centrifuged and the supernatant was removed. After a wash step, the cells were stained with antibodies against CD45 APC-Conjugated antibody (Miltenyi Biotec) and acquired by FACSCanto™ Cytometer (BD, Franklin Lakes, NJ, USA). Monocytes and granulocytes were gated, through their morphological characteristics, to assess bioparticle internalization, as shown in Figure 4A. All data were analyzed with Kaluza Analysis 2.0 software (Beckman Coulter, Indianapolis, IN, USA). 

### 2.5. Nuclear Translocation of NF-κB

To assess nuclear translocation of NF-κB, human monocytes were purified and stimulated with LPS O1, LPS O2a and LPS O2afg for 6 h as described above. Inhibition with PMB was done also as mentioned above. Cells were fixed with 4% para-formaldehyde for 10 min, permeabilized with 0.1% Triton/PBS for 10 min and blocked with 1% BSA/PBS 1% for 1 h at room temperature. Immunofluorescence was done with a primary rabbit antibody against NF-κB/p65 (sc-372; Santa Cruz Biotechnology, Dallas, TX, USA) for 1 h and a FITC-conjugated donkey anti-rabbit secondary antibody (Jackson ImmunoResearch, Suffolk, UK) for 45 min. Nuclei were stained with DAPI. Samples were visualized under a Leica confocal station (Leica SP5 confocal system) mounted on a Leica DM6000 inverted microscope (Leica Microsystems Inc., Buffalo Grove, IL, USA). Areas of NF-κB were quantified using ImageJ 1.53h software (National Institutes of Health, Be- thesda, MD, USA).

### 2.6. Real-Time PCR

Real-time PCR for assessing bacterial genes transcription was performed as mentioned elsewhere [32]. Total *K. pneumoniae* RNA was extracted with the RNeasy Mini Kit (Qiagen) after lysis with Pathogen Lysis Tubes S (Qiagen). Subsequently, 500 ng of RNA was reverse-transcribed with the High Capacity RNA-to-cDNA kit (Thermo Fisher Scientific) and used as a template in a real-time PCR with a SYBR Select Master Mix (Thermo Fisher Scientific) and a specific primer pair for the following genes fimA (F- TGGATGATTGCGACACTACG; R- TATTGTCGAGGATCTGCACG), fimH (F- GACCAACAACTACAATAGCGAC; R- ATTGGTGAAGATCGCGTTGG), ompK36 (F- ACGCGGGCTCTTTCGACTAC; R- AGTTGTCAGAACCGTAGGTG) and rho (F- AACTACGACAAGCCGGAAAA; R- ACCGTTACCACGCTCCATAC) was used as the housekeeping gene for the relative quantification [33], assessed by 2-ΔΔCT calculation for each mRNA [34].

### 2.7. Statistical Analysis

Statistical analyses were performed using the Mann–Whitney U test (ELISA and RT-PCR assays) and one-way ANOVA followed by Dunnett’s test when comparing more than two groups or two-way analysis of variance (ANOVA) for multiple comparisons followed by Bonferroni’s test (whole blood, monocytes and phagocytosis). In some cases, a Student’s *t*-test was used to evaluate the significance against the hypothetical zero value. Experiments were replicated at least three times. All statistical analyses were performed using GraphPad Prism 8.0 software (San Diego, CA, USA). The results obtained were expressed as the mean ± SEM. Data were considered statistically significant when a value of *p* < 0.05 was achieved.

### 2.8. Ethical Clearance

Informed consent was obtained from all IRCCS-ISMETT patients whose clinical sera samples were used in this study, under internal protocol IRBB/30/20, for LPS ELISA assays and immunoblots. For all other experiments described in this work, human blood from healthy individuals (20 to 50 years old) was used. All healthy volunteers gave oral and written consent.

## 3. Results

### 3.1. Determination of O-Antigen Serotype of IRCCS-ISMETT Clinical Samples

The genetic analysis of 178 IRCCS-ISMETT *K. pneumoniae* clinical isolates showed that they belong to several STs, including 37 ST258, 28 ST307 and 51 ST512. In Italy, CRE *K. pneumoniae* strains are predominantly one of these three STs [35,36]. Out of 178 *K. pneumoniae* isolates, 163 belonged to the O2afg serotype, i.e., 91.6%. Other O-antigens identified were O1, O2a, O3, O4 and O12. Concerning ST258, ST307 and ST512 clinical samples, 97.3%, 96.4% and 98.0% of these isolates possess the O2afg-antigen, respectively.

### 3.2. Assessment of the Presence of Antibodies against LPS Serotypes O1, O2a and O2afg in Human Sera from Patients Who Suffered Either an Infection or a Colonization Case of K. pneumoniae

We next sought to confirm if O2a and O2afg antigens are less immunogenic than O1-antigen in our in-house sera samples, as is previously observed in other health structures [16]. Using sera from in-house infection and colonization cases, we observed, as expected, that antibodies against LPS O1 have a significant higher titer than when compared to antibodies against both LPS O2a and LPS O2afg (Figure 1A). In order to ensure that the differences observed were not due to different LPS glycans’ affinity for the ELISA plate material, we also performed a dot blot by loading the different LPS in a membrane and incubating them with a dilution 1:1000 of sera of an infected patient, a colonization case and a negative individual (Figure 1B). As expected, the antibodies present in the sera of the infected patient recognized all the quantities of LPS tested. On the other hand, antibodies against the different LPS in the colonization case barely recognized the lowest LPS quantity used (0.125 µg), except for the ST258 O2afg LPS. Unsurprisingly, the negative serum failed to recognize all of the LPS at all quantities used (Figure 1B).

### 3.3. Quantification of Pro-Inflammatory Chemokines and Cytokines Induced by LPS O1, LPS O2a or LPS O2afg

Since monocytes have an important role in modulating the immune response during a bacterial infection, we then assessed the levels of pro-inflammatory cytokines and chemokines secreted by these cells when incubated with the purified and different *K. pneumoniae* LPS. After 48 h, we collected the supernatants to evaluate the secretion of pro-inflammatory factors. Among the 30 inflammatory cytokines and chemokines tested, we observed differences in the levels of TNF-α, IL-6, IL-12p40, MIP-1α and MCP-1 (Figure 2A–E). All the other cytokines and chemokines tested showed no significant differences (Appendix A).

At 48 h, an increase in TNF-α, IL-6, IL-12p40, MIP1α and MCP-1 levels was observed only with the LPS O1-antigen (Figure 2A–E). An increase in IL-12p40 was observed for LPS O2afg-antigen from strain 6613 (Figure 2C). Intrigued with this result, we measured the same factors after 6 h, to understand how the different LPS affected those inflammatory levels at an early time point. We observed all LPS serotypes induced an initial inflammatory response at 6 h (Figure 2F–L) which eventually increased when monocytes were stimulated with LPS O1 but it is short-lived when stimulation was done with LPS O2a and LPS O2afg. As before, no alteration in the levels of other tested immunological factors was observed (Appendix A). Also, the inflammatory potential of all tested LPS antigens was inhibited by pre-incubating monocytes with PMB, a cationic peptide with strong antibiotic properties, which binds and neutralizes LPS (Figure 2F–L). Interestingly, after 6 h, monocytes triggered with LPS O2afg from strain 6613 did not produce any significant pro-inflammatory molecules. We performed the same experiment using whole blood cells, a more complex model of inflammation (Figure 3A–E), in which both LPS O1 and LPS O2a induce secretion of high levels of inflammatory cytokines (except for IL-6 in cells stimulated with LPS O1) whereas, cells treated with LPS O2afg serotypes (6613 and ST258) induce low to none cytokines secretion (except for MCP-1 and IL-1β). LPS O2a stimulates high levels of TNF-α and IL-6 (Figure 3A,B).

Interestingly, all LPS antigens induced high levels of MCP-1 (Figure 3D). In these conditions, as observed previously for monocytes, the levels of other cytokines and chemokines remained unchanged (Appendix A). As before, PMB contrasted the effect of the different LPS on TNF-α, IL-6, IL-12p40, MIP1α and MCP-1 levels for LPS O1-antigens and LPS O2a-antigens (Figure 3A–E) and, importantly, it did not affect cell viability (data not shown). Moreover, preliminary experiments with commercial *K. pneumoniae* O1-serotype led to similar results (data not shown). However, after observing that this particular LPS was very impure, which is likely due to contamination of capsular polysaccharides (Appendix A), we could not exclude the possibility of the induction of immunological responses caused by the latter type of polysaccharides that could mask the actual one caused by the LPS we were studying and the decision not include the commercial LPS in further experiments was made.

### 3.4. Phagocytosis of K. pneumoniae Strains Expressing LPS O1, LPS O2a or LPS O2afg

In order confirm if the different O-antigens have a role in bacterial phagocytosis, we performed a phagocytic assay using monocytes and granulocytes from whole blood samples and *K. pneumoniae* strains displaying the relevant LPS O-antigens (B5055 for O1, C5046 for O2a and 6613 and ST258 for O2afg). The level of phagocytic activity was then measured by flow cytometry (Figure 4A). No significant differences were observed for all bacterial strains used, except for strain 6613, which was eagerly phagocyted by both monocytes and granulocytes (Figure 4B,C).

### 3.5. Nuclear Translocation of NF-κB to the Nuclei of Human Monocytes Stimulated with K. pneumoniae LPS

One of the most important mediators of LPS response is NF-κB. Thus, we evaluated if, in monocytes stimulated with the *K. pneumoniae* LPS, NF-κB nuclear translocation would occur. We observed that NF-κB is translocated to the nuclei in monocytes incubated with E. coli LPS, O1-antigen and, to a lesser extent, with LPS O2a (Appendix A). No translocation was observed in cells treated with LPS O2afg-antigen. As expected, treatment with PMB reversed the LPS-mediated translocation of NF-κB to the nuclei (Appendix A).

### 3.6. Transcriptional Levels of K. pneumoniae Genes Related with Phagocytosis

Previous work has shown that some *K. pneumoniae* proteins can either protect against or promote phagocytosis [2]. *K. pneumoniae* strains expressing type 1 fimbriae proteins FimA and FimH are more prone to be phagocyted and strains expressing the membrane protein OmpK36 are more resistant to phagocytosis [2]. We quantified the level of transcription of bacterial genes encoding for such proteins in the strains used in this work, and observed no significant differences (Figure 5).

## 4. Discussion

In order to successfully colonize and, subsequently infect, *K. pneumoniae* must evade host immune mechanisms. As an example, it can easily avoid the complement system and some strains are resistant to phagocytosis [2,12]. *K. pneumoniae* can affect innate immune pathways by inhibiting certain pathways, e.g., NF-κB or TLR pathways. Capsular polysaccharides also play a role, since non-capsulated mutants are less virulent and less prone to disseminate in vivo. It has been observed that LPS protects *K. pneumoniae* from complement by hindering the deposition of C3b in the bacterial outer membrane [37]. Lipid A can also undergo structural modifications on its side chains, which provides resistance to host cationic antimicrobial peptides (CAMPs), thus rendering *K. pneumoniae* more virulent [38]. On the other hand, LPS O-antigen seems to interfere in the TLR2-TLR4-MyD88 inflammation pathway since mutants lacking the O-antigen induce higher levels of inflammation through this pathway [39]. However, different *K. pneumoniae* phenotypes (hypervirulent, several types of multidrug resistance, drug sensitive, non-capsulated mutants, etc.) and origins (nosocomial, environmental, etc.) exhibit diverse mechanisms of immune evasion [2,12]. Particular STs, such as ST258 and ST512, show an enormous plastic ability to acquire resistance genes, which renders them successful nosocomial pathogens [2]. However, the link between acquisition of antibiotic resistance and poor immune recognition of O2afg is not straightforward since strains from the same ST can display different O-antigens. We believe that those expressing O2afg tend to prevail partly due to the contribution of this particular O-antigen in inducing poor immune responses. While O1-antigen predominates in susceptible *K. pneumoniae* isolates, O2afg-antigens are more frequently observed in CRE strains [16,17]. We observed a similar tendency with our in-house CRE clinical isolates, where more than 91% belonged to the O2afg serotype. Interestingly, most of our O2afg-antigen clinical isolates belong to STs of clinical relevance, such as ST258, ST307 and ST512. However, some exceptions were observed. We recently characterized CRE *K. pneumoniae* ST392 clinical isolates in terms of virulence and observed that they have the potential to become an ST of clinical relevance [32]. Despite their CRE phenotype, ST392 isolates expressed O4-antigen. Thus, CRE strains expressing O-antigens other than O2a or O2afg either share their immunomodulatory properties or rely on other immune evasion mechanisms that are mediated by either their LPS or other virulence factors.

LPS O1-antigen tends to induce a higher humoral immune response than the LPS O2-antigen [16,17]. Using sera from patients who were diagnosed either with an infection or a colonization case by *K. pneumoniae*, we confirmed that tendency (Figure 1A). Monoclonal antibodies against O2-antigen are protective in mouse models of pneumonia and sepsis [16]. Thus, due to the phagocytic nature of antibodies against O2-antigen, which is likely extended to antibodies against O2a and O2afg, the low prevalence of these antibodies would favor the bacterial dissemination of multidrug resistant *K. pneumoniae* strains. Interestingly, we observed that *K. pneumoniae* LPS O1-antigen and, to a lesser extent, LPS O2a-antigen induce sustained cellular inflammatory responses. On the other hand, LPS O2afg-antigens induced a short-lived inflammatory response. We have also seen that secretion of MCP-1 and IL-1β is induced by all LPS at an early stage. Since MCP-1 is a monocyte-chemoattractant, *K. pneumoniae* strains expressing these LPS O-antigens would induce the migration of monocytes and other immune cells to the site of infection. Human monocytes stimulated with *E. coli* LPS secrete high levels of MCP-1 and MIP-1α in a time-dependent fashion [40]. Likewise, the release of these two cytokines behaved similarly when monocytes were incubated with the *K. pneumoniae* LPS O1-antigen (Figure 2D,E,I,L). However, in agreement with a potential immune evasion mechanism, MCP-1 secreted by monocytes stimulated with LPS O2afg-antigen from an ST258 clinical isolate peaks earlier than the control LPS and LPS O1-antigen at 6 h (Figure 2L), but it is short-lived since, at 48 h, levels of MCP-1 induced by LPS O2afg-antigen from an ST258 strain are practically null (Figure 2E). This would contribute to an impaired recruitment of monocytes to the site of infection and allow bacterial dissemination of clinically relevant O2afg-serotype *K. pneumoniae* strains. Interestingly, LPS O2afg-antigen derived from an ST258 strain induced higher levels of all secreted cytokines (Figure 2F–L) in monocytes when compared to those elicited by the LPS O2afg-antigen purified from strain 6613. Although these molecules belong to the same O-serotype, one may not exclude the structural differences in the remaining components of the LPS molecule, such as lipid A and core oligosaccharide [2]. Both regions and their respective structural differences have been shown to differently affect inflammatory responses in murine and human monocytes [2,41]. Different O-antigens possess different sugar compositions [14] and we believe that the different O-antigens affect the inability of O2afg to proper stimulate an inflammatory response. When the sequence of the enzyme responsible for modifications in the structure of Lipid A is observed, the aspartyl beta-hydroxylase LpxO protein is 100% identical for all the reference strains and the ST258 clinical isolate used in this work, which enables our belief that the sequence of lipid A does not vary between the different purified LPS used in this work. However, the structural differences in the core oligosaccharide of the LPS derived from the ST258 and 6613 strain may still result in the inflammatory differences observed by us.

LPS O1-antigens and, to a lesser degree, O2-antigens activate the inflammatory pathways resulting in NF-κB translocation to cell nuclei. This process controls cytokines production when stimulated by different factors, such as bacterial or viral infections, and is dependent of several factors, such as IL-1 β, TLR activation by LPS and TNF-α. Although all LPS induced the production of IL-1β, nuclear translocation of NF-κB also depends on TNF-α and the binding of LPS to TLR such that IκB kinases can be inactivated [42]; this allows the migration of NF-κB to the nuclei [39]. In agreement, our preliminary results reflect the inability of O2afg-antigen to both efficiently stimulate TNF-α and to properly bind to TLR, which may impair the synergetic activation and consequent migration of NF-κB to the nuclei (Appendix A). In agreement, PMB abolishes the LPS O1 and LPS 02a NF-κB nuclear translocation (Appendix A), as expected [43]. Moreover, we showed that the 6613 strain is eagerly phagocyted (Figure 4A–C). Combined with other factors such as capsular polysaccharides, the ability of LPS O2afg of 6613 strain to induce IL-1β secretion may initiate the differentiation of monocytes into macrophages that express phagocytic markers such as CD16, CD36, CD64 and CD206 through the TLR2/1 pathway [44], resulting in the high levels of phagocytosis observed for this particular *K. pneumoniae* strain. In agreement, we observed NF-κB translocation to the cell nucleus in LPS O1-antigen and LPS O2a-antigen stimulated monocytes, which was reverted by pre-incubation with PMB (Appendix A). However, it has been shown that NF-κB shuttles back and forth from the nucleus to the cytoplasm following stimulation with LPS [45]. The translocation of NF-κB monomers other than p65, such as p50, and their consequent dimerization may also activate the transcription of pro-inflammatory molecules [46]. In addition, p65 is a crucial factor of the NF-κB ATF3-CEBPδ transcriptional pathway, which modulates the transition between transient to persistent and vice-versa of TLR4 stimulation in macrophages [46]. In order to fully understand the effect of *K. pneumoniae* LPS O2afg-antigen in modulating immune responses through NF-κB, these factors must also be considered and future studies will unravel their role in this immune evasion mechanism. When monocytes’ stimulation is performed with the LPS O2afg-antigen (either from 6613 or ST258), PMB seems to restore the levels of pro-inflammatory molecules (Figure 3B,C,E), which suggests that PMB could not only act as a negative immune-modulator of inflammatory response, but, at an early stage, it could revert immune evasion as previously demonstrated for other bacterial LPS [47,48,49,50].

The ability to avoid phagocytosis seems to be LPS-unrelated. In fact, the only strain that was eagerly phagocyted was 6613, which possesses the same LPS O2afg-antigen than the ST258 strain (Figure 4B,C). Thus, LPS O2afg-antigen does not play a role in protection against phagocytosis. *K. pneumoniae* strains expressing type 1 fimbriae proteins FimA and FimH are more prone to be phagocyted [2] and strains expressing the membrane protein OmpK36 are more resistant to phagocytosis [2]. However, no significant differences were observed in the transcription levels of these molecules by the strains used in this work (Figure 5A–C). As stated elsewhere, capsular polysaccharides have an important role in the protection against phagocytosis [2]. Although 6613 and ST258 strains share the same O2afg-antigen, they possess different capsular loci (KL27 and KL107, respectively). In agreement, ST258 strains are more resistant to phagocytosis in a murine model [2] and B5055 and C5046 also possess capsular loci (KL2 and KL3, respectively) associated with protection against the phagocytic processes [2,51].

Our results show that different *K. pneumoniae* LPS O-antigens induce diverse inflammatory responses ex vivo. This suggests that multidrug resistant *K. pneumoniae* strains take advantage of the poor ability of their O2afg-antigens to induce inflammatory responses to survive and disseminate within the host; this causes the pathologies related to this bacterial pathogen.

## Figures and Tables

**Figure 1 microorganisms-09-01317-f001:**
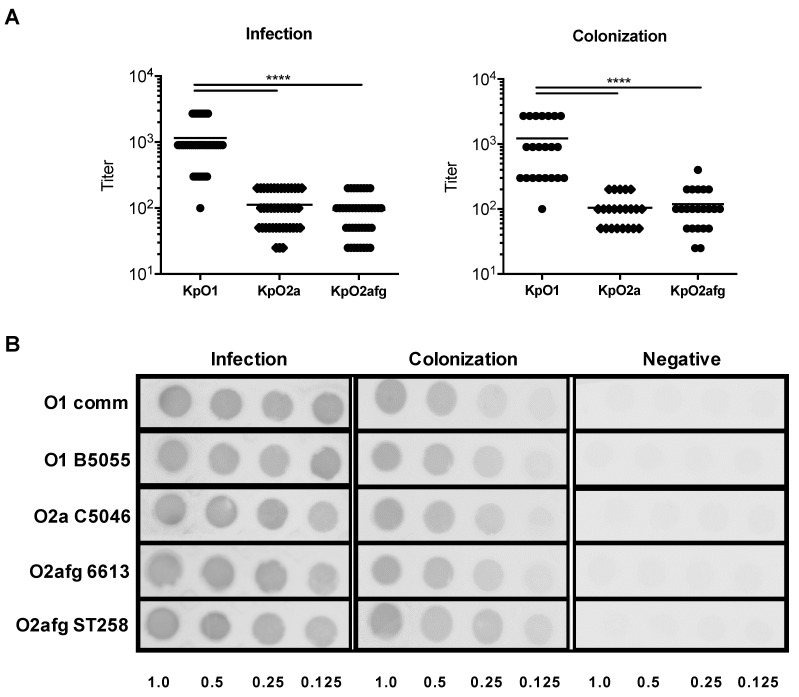
Following *K. pneumoniae* infection or colonization, titers of the antibodies against LPS O1, O2a and O2afg present in human patients’ sera were determined in infection and in colonization cases (**A**). Statistical analysis was performed using the Mann–Whitney U test, **** *p* < 0.0001. Immunoblots used different LPS quantities (1, 0.5, 0.25 and 0.125 µg) and sera from an infection case, a colonization patient and a negative individual at a dilution of 1:1000 (**B**).

**Figure 2 microorganisms-09-01317-f002:**
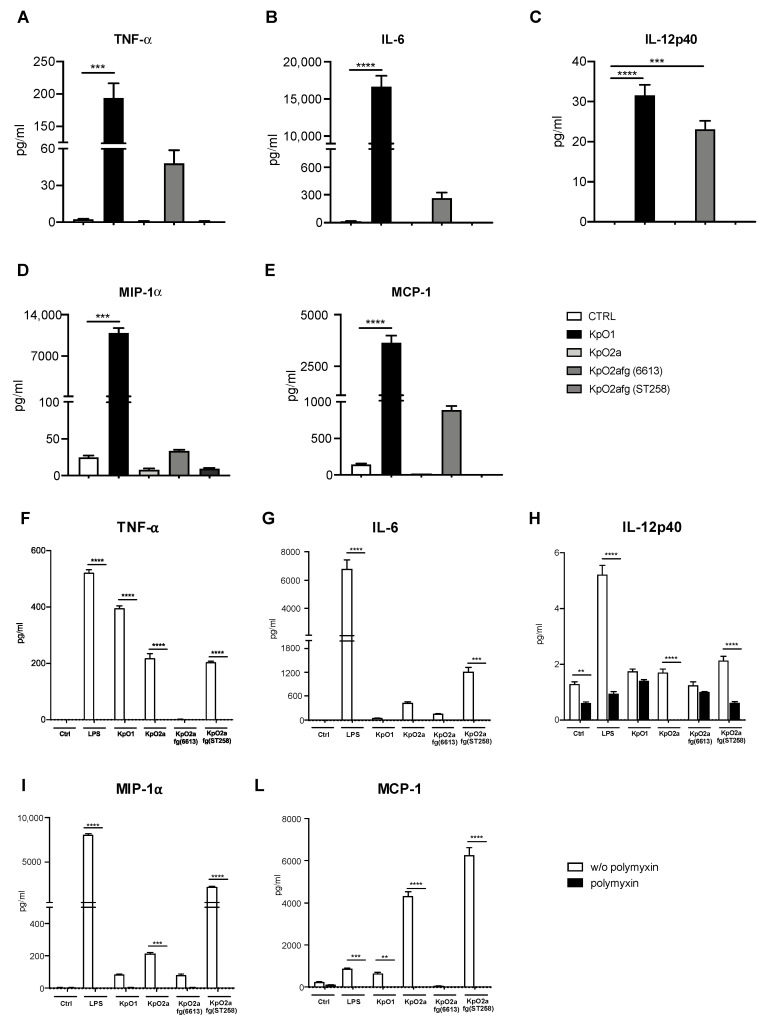
The stimulation of isolated monocytes for 48 h (**A**–**E**) and 6 h (**F**–**L**) with 1 µg/mL of each of the different *K. pneumoniae* purified LPS (KpO1, KpO2, KpO2afg (6613) and KpO2afg (ST258)). LPS refers to E. coli 0111:B4 commercial LPS and it was also used at a concentration of 1 µg/mL. Results are representative of four different donors. The results are expressed as the average ± SEM. Statistical analysis was performed using one-way ANOVA, ** *p* < 0.01, *** *p* < 0.001 and **** *p* < 0.0001.

**Figure 3 microorganisms-09-01317-f003:**
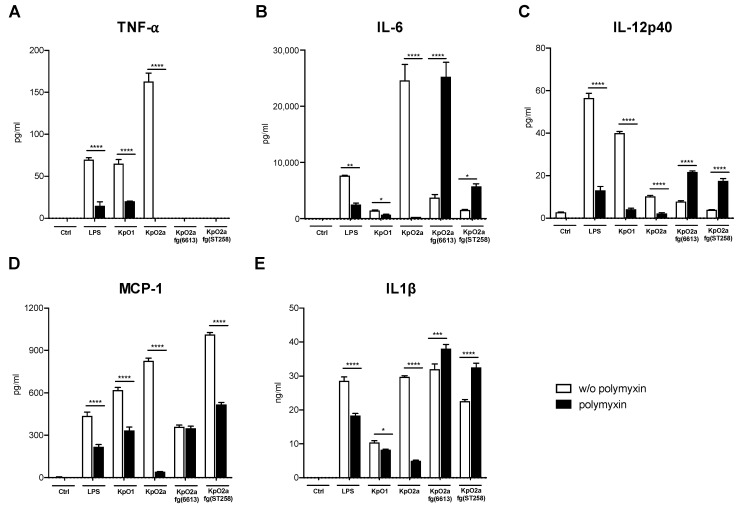
Whole blood cells were stimulated with 1 µg/mL of each of the *K. pneumoniae* purified LPS KpO1, KpO2, KpO2afg (6613) and KpO2afg (ST258) for 6 h (**A**–**E**). LPS refers to E. coli 0111:B4 commercial LPS used at the same concentration as the *K. pneumoniae* LPS. Results are representative of four different donors. The results are expressed as the average ± SEM. Statistical analysis was performed using one-way ANOVA, * *p* < 0.05, ** *p* < 0.01, *** *p* < 0.001 and **** *p* < 0.0001.

**Figure 4 microorganisms-09-01317-f004:**
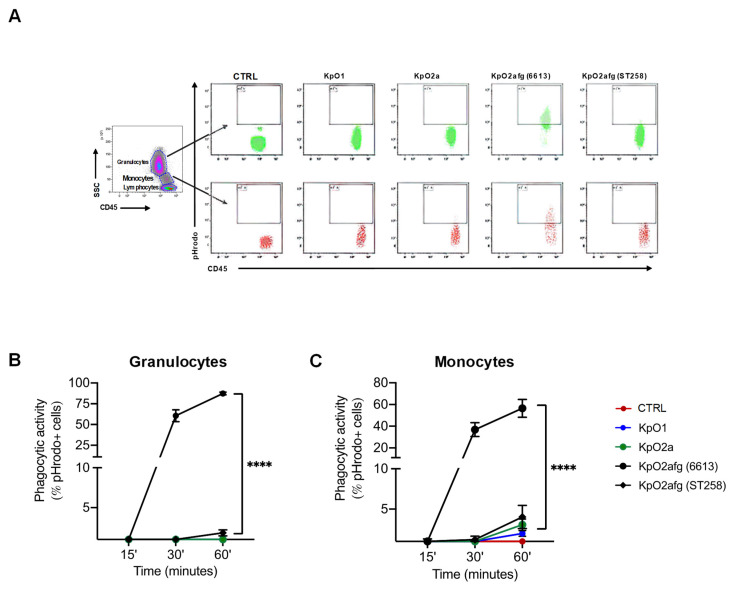
Phagocytosis of *K. pneumoniae* strains (KpO1, KpO2, KpO2afg (6613) and KpO2afg (ST258)) in peripheral whole blood. (**A**) Flow cytometry plots with gating strategy and phagocytic activity levels of (**B**) granulocytes and (**C**) monocytes are shown. Phagocytosis was measured after 60 min of incubation with stained bacteria using the pHrodo™ Red Phagocytosis Particle labelling kit. CTRL refers to the control used, i.e., phagocytosis blocked by incubation in ice for 60 min, and is representative of what was observed for all controls of the *K. pneumoniae* strains used in this experiments. Statistical analysis was done using ordinary one-way ANOVA, **** *p* < 0.0001.

**Figure 5 microorganisms-09-01317-f005:**
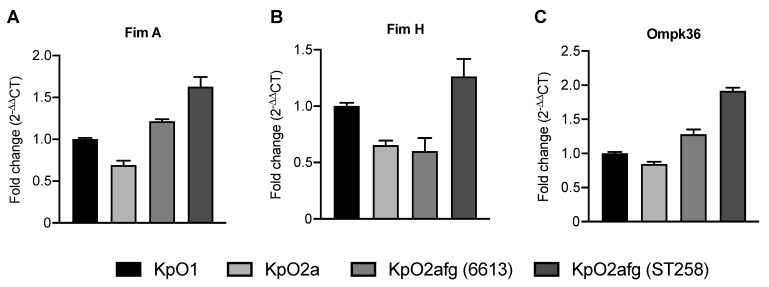
Transcriptional levels of fimA (**A**), fimH (**B**) and ompK36 (**C**). *K. pneumoniae* mRNA from strains B5055, C5046, 6613 and St258 were quantified by Real-Time PCR. *rho* was used as a reference gene for the relative quantification and assessed by 2^−ΔΔCT^ calculation for each mRNA. Fold changes were calculated based on B5055 transcriptional levels.

## Data Availability

The datasets generated for this study can be found in the BioProject ID PRJNA687927 (https://www.ncbi.nlm.nih.gov/bioproject/?term=PRJNA687927, accessed on 2 November 2020).

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
