# Peer review of "Klebsiella pneumoniae* Lipopolysaccharides Serotype O2afg Induce Poor Inflammatory Immune Responses Ex Vivo"

_microorganisms, 2021, doi:10.3390/microorganisms9061317_

Round 1

Reviewer 1 Report

I thoroughly enjoyed reading this manuscript as I personally think that we have not fully explored and appreciated the wide range of inflammatory responses induced by different types of bacterial pathogens. As the authors rightly highlighted it, this is (now more than ever) a very important area of translational research as antibiotic resistance is indeed an world-wide emergency.

The study is simple and straightforward in its design - yet very intriguing for the answers it has provided.

I have listed here below a few questions/curiosities that I would appreciate if the authors could answer:

The data in Figure 2S seem to be pivotal for the interpretation of the results. I am not sure they should be supplementary. This is author's choice of course but I would suggest they go as the main figure;

One general comment is that the results are not fully commented on. As an example, Figure 1L: why is ST258 more potent than LPS in inducing MCP-1? Similarly, Figure 2: why is polymyxin B increasing (2B, 2C, 2E) the release of some cytokines when added to the tested antigens? I think that these are very exciting results that the authors seem to brushing under the carpet;

The text of the figure legends have comments that should be better placed in the Results section;

This is again my personal comment but I would remove the NF-kB data altogether. First and foremost, the translocation of NF-kB at a given time is conceptually not very useful as several studies have shown that NF-kB shuttles back and forth between nucleus and cytoplasm multiple times even within few hours from stimulation. Therefore, it is the dynamic of shuttling rather than the single-shot picture of NF-kB localisation that counts. This is specifically true for LPS stimulation (requiring both IkBalpha and beta degradation) and innate cells. 

Author Response

#Reviewer 1

I thoroughly enjoyed reading this manuscript as I personally think that we have not fully explored and appreciated the wide range of inflammatory responses induced by different types of bacterial pathogens. As the authors rightly highlighted it, this is (now more than ever) a very important area of translational research as antibiotic resistance is indeed an world-wide emergency.

The study is simple and straightforward in its design - yet very intriguing for the answers it has provided.

We thank the reviewer for his/her encouraging words.

I have listed here below a few questions/curiosities that I would appreciate if the authors could answer:

1 - The data in Figure 2S seem to be pivotal for the interpretation of the results. I am not sure they should be supplementary. This is author's choice of course but I would suggest they go as the main figure;

We agree with the reviewer and we have now moved this figure to the main manuscript.

2 - One general comment is that the results are not fully commented on. As an example, Figure 1L: why is ST258 more potent than LPS in inducing MCP-1? Similarly, Figure 2: why is polymyxin B increasing (2B, 2C, 2E) the release of some cytokines when added to the tested antigens? I think that these are very exciting results that the authors seem to brushing under the carpet;

We thank the reviewer for his/her suggestion and we have now added to the Discussion the following: “Human monocytes stimulated with E. coli LPS secrete high levels of MCP-1 and MIP-1α in a time-dependent fashion [39]. Likewise, release of these two cytokines behaved similarly when monocytes were incubated with K. pneumoniae LPS O1-antigen (Fig. 1D,E,I,L). However, and in agreement with a potential immune evasion mechanism, MCP-1 secreted by monocytes stimulated with LPS O2afg-antigen from an ST258 clin-ical isolate peaks earlier than the control LPS and LPS O1-antigen at 6 h (Fig. 2L), but it is short-lived, since at 48 h, levels of MCP-1 induced by LPS O2afg-antigen from ST258 strain are practically null (Fig. 2E). This would contribute to an impaired recruitment of monocytes to the site of infection and allow bacterial dissemination of clinically relevant O2afg-serotype K. pneumoniae clinical strains. Interestingly, LPS O2afg-antigen derived from an ST258 strain induces higher levels of all secreted cytokines (Fig. 1 F-L) in monocytes when compared to those elicited by LPS O2afg-antigen purified from strain 6613. Although these molecules belong to the same O-serotype, one may not exclude structural differences in the remaining components of the LPS molecule, such as lipid A and core oligosaccharide [2]. Both regions and their respective structural differences have been shown to affect differently inflammatory responses in murine and human monocytes [2,40]. Different O-antigens possess different sugar compositions [14] and we believe that affects the inability of O2afg to proper stimulate an inflammatory re-sponse. When observed that the sequence of the enzyme responsible for modifications in the structure of Lipid A, the aspartyl beta-hydroxylase LpxO protein, is 100% identical for all the reference strains and ST258 clinical isolate used in this work, leading to be-lieve that the sequence of lipid A does not vary between the different purified LPS used in this work. However, structural differences in the core oligosaccharide of the LPS de-rived from ST258 and 6613 strain still may result in the inflammatory differences ob-served by us.”.

We respectfully note we have commented the effects of polymyxin B in the increase of the release of some cytokines already in the previous version of the Discussion, where we wrote: “. When monocytes’ stimulation is done with the O2afg-antigen (either from 6613 or ST258), PMB seems to restore the levels of pro-inflammatory molecules (Fig. 3B,C,E), suggesting that PMB could not only act as a negative immune-modulator of inflam-matory response, but, at an early stage, it could revert immune evasion, as previously demonstrated for other bacterial LPS [44–47].”.

3 - The text of the figure legends have comments that should be better placed in the Results section;

We agree with the reviewer and we have now moved those comments from the figures legends to the Results section.

4 - This is again my personal comment but I would remove the NF-kB data altogether. First and foremost, the translocation of NF-kB at a given time is conceptually not very useful as several studies have shown that NF-kB shuttles back and forth between nucleus and cytoplasm multiple times even within few hours from stimulation. Therefore, it is the dynamic of shuttling rather than the single-shot picture of NF-kB localisation that counts. This is specifically true for LPS stimulation (requiring both IkBalpha and beta degradation) and innate cells.

We believe that the reviewer’s suggestion is adequate and we have moved the NF-κB data to the supplementary data and just briefly mention it in the Discussion as preliminary data due to the shuttling of this molecule back and forth from the nucleus following LPS stimulation.

Reviewer 2 Report

Dear Author/s,

Submitted manuscript microorganisms-1228089 has poorly written, every section of the paper needs to revise. There are four places where you have written “Data not shown”, but it is needed to show here, at least two places, or else in supplementary document.

Lipopolysaccharides (LPS) from Klebsiella pneumoniae (Product Number L 1519) from Sigma should be used as control in all experiments, there is no use of LPS extracted from E. coli as biological/experimental control here.

Methodology is repetitive, example sections 2.2 have same information given in 2.1; Similarly, 2.8 and supplementary figure S3 legends are same. Flow cytometry of Phagocytosis assay in whole blood (ex vivo) is incomplete, use of antibody and removal of other blood cells etc. not being mentioned anywhere.

That many sections of methods (2.1 to 2.11) and only 4 results sections (3.1-3.4- not described well) with only 3 figures and at 4 places “Data not shown”, how can one validate the findings?

The list of such instances is too lengthy and not easy to point out every pitfall here. It is author/s duty at the time of submission, take care of all these shortcomings.

All the best.

Author Response

#Reviewer 2

1 - Submitted manuscript microorganisms-1228089 has poorly written, every section of the paper needs to revise.

We respectfully disagree with the reviewer. We and the 2 other reviewers, based on their comments, do not feel the manuscript is poorly written, even after we revised the entire manuscript.

2 - There are four places where you have written “Data not shown”, but it is needed to show here, at least two places, or else in supplementary document.

We agree with the reviewer that the “data not shown” were redundant and removed most of them, as the information was well described in the text. We just left two “data not shown”, one concerning the cell viability, and the other the use of impure K. pneumoniae commercial LPS, since this information would not add anything new or support the mainstream data.

3 – Lipopolysaccharides (LPS) from Klebsiella pneumoniae (Product Number L 1519) from Sigma should be used as control in all experiments, there is no use of LPS extracted from E. coli as biological/experimental control here.

We understand the reviewer’s concern and we agree with it. However, after performing the HPLC experiments, we observed that commercial K. pneumoniae LPS O1 (L4268, also from Sigma) contains many impurities, most probably derived from capsular polysaccharides. Conversely, our purified LPS O1 is much purer. Since capsular polysaccharides also induce the secretion of pro-inflammatory cytokines (reviewed in Paczosa & Mecsas, Microbiol Mol Biol Rev, 2016), we believe that the commercial LPS O1 would stimulate monocytes and whole blood cells differently than our purified LPS O1 and, therefore, they cannot be compared.

4 – Methodology is repetitive, example sections 2.2 have same information given in 2.1; Similarly, 2.8 and supplementary figure S3 legends are same. Flow cytometry of Phagocytosis assay in whole blood (ex vivo) is incomplete, use of antibody and removal of other blood cells etc. not being mentioned anywhere.

We thank the reviewer for having noticed that. We have now rewritten and reduced the Methods section. Regarding the phagocytosis protocol, we respectfully note that it includes all the information required by the reviewer “Labelling was done using the pHrodo™ Red Phagocytosis Particle Labeling Kit for Flow Cytometry (Thermo Fisher Scientific) as for manufacturer’s instructions. The phagocytosis assay was performed by exposing the heparinized whole blood sam-ple to pHrodo™ dye-labeled bacteria for 60 min at 37°C. White blood cells were stained with CD45 APC-Conjugated antibody (Miltenyi Biotec), acquired by FACSCanto™ Cytometer (BD) and analyzed with the Kaluza Analysis 2.0 software (Beckman Coulter, Indianapolis, IN, USA).” The removal of erythrocytes is done using the buffers that are part of the mentioned kit, and the overall protocol is the one suggested by the kit’s manufacturer.

5 – That many sections of methods (2.1 to 2.11) and only 4 results sections (3.1-3.4- not described well) with only 3 figures and at 4 places “Data not shown”, how can one validate the findings?

We thank the reviewer for his/her suggestion and, as mentioned above, we have rewritten these entire sections. We hope they will be, in their current form, more clear to the readers.

6 – The list of such instances is too lengthy and not easy to point out every pitfall here. It is author/s duty at the time of submission, take care of all these shortcomings.

Again, we respectfully disagree with the reviewer, since the other reviewers did not mention so many pitfalls or pointed out criticisms at such level.

Reviewer 3 Report

In the present study entitled „Klebsiella pneumoniae lipopolysaccharides serotype O2afg induce poor inflammatory immune responses ex vivo” authors have shown the effects of different LPS serotypes of K. pneumonie on human monocytes. The study is original and novel in understanding the inflammatory immune responses against K. pneumonie infections. The experiments appeared to be well conducted, however, there are some issues reqiured to be addressed:

  1. Introduction is quite short and maybe it would be useful to expend it with additional information about the action of K. Pneumoniae LPS on macrophages and moreover on the pro-inflammatory cytokine production.
  2. Material and methods: Please give a brief description of NGS in section 2.2. Please describe briefly the cell viability assay used in the experiments in section 2.5. Please describe briefly the isolation of the monocytes in section 2.6. Please give the exact serotype of E. coli LPS used as controls.
  1. The authors described the translocation of the p65 protein of the NFkB pathway. The translocation of p50 would be also informative since p50 in homodimers ia also able to activate the transcription of the pro-inflammatory cytokines. Did the autohors consider that other signalling pathways e.g. C/EBPβ may contribute to cytokine production?
  2. On which basis did the authors used 1 ug/mL of LPS at the treatments? Did the authors carry out concentration dependence experiments?
  3. How the authors explain that the IL-1β level increased in case of O2afg, meanwhile the NFkB translocation decreased?
  4. The IL-1β level increased in case of O2afg as well as the phagocytic activity of monocytes. Is it possible to have connection between them? Please discuss this issue in the manucript.
  5. Is it possible that the different O antigens act on the inflammasome? Please discuss this in the manucript.
  6. Point and comma are equally used for the decimal signs e.g lines 134; 217; 219; 232, versus lines 115, 118, 119; 125.
  7. Figures (Figure 1-3) are needed to be reconsidered, since the font size is very small, and maybe only in the pdf version, but they are in vertical form. Please re-organize the structure of the figures because the current form is not friendly for readers.

Author Response

#Reviewer 3

In the present study entitled „Klebsiella pneumoniae lipopolysaccharides serotype O2afg induce poor inflammatory immune responses ex vivo” authors have shown the effects of different LPS serotypes of K. pneumonie on human monocytes. The study is original and novel in understanding the inflammatory immune responses against K. pneumonie infections. The experiments appeared to be well conducted, however, there are some issues reqiured to be addressed:

1 – Introduction is quite short and maybe it would be useful to expend it with additional information about the action of K. Pneumoniae LPS on macrophages and moreover on the pro-inflammatory cytokine production.

We agree with the reviewer’s suggestion and we have now modified the Introduction to include more information about the action of K. pneumoniae LPS on macrophages, which now reads: “Lipopolysaccharides (LPS) are components of the bacterial outer membrane composed by an O-antigen, a core oligosaccharide and lipid A. Many pathogens use their capsular polysaccharides and structurally-modified LPS to escape host microbicidal mechanisms such as phagocytic and macrophage action [7]. As an example, K. pneumoniae can survive within macrophages due to its capsular polysaccharides [8] while downregulation of the latter increases bacterial phagocytosis by macrophages [9] and some host factors also downplay LPS protective action, such as CD36 [10]. During K. pneumoniae infections, LPS O-antigens can bind the complement component C3b, impairing complement-mediate killing and promoting bacterial survival [2]. However, LPS can induce inflammation through the binding of lipid A to TLR4 and consequent initiation of an inflammatory cascade of production of chemokines and cytokines to fight bacterial infection [2,11,12]. Interestingly, different K. pneumoniae strains can induce immune response with diverse outcomes. A previous work using CRE K. pneumoniae strains A28006 and A54970, be-longing respectively to ST11 and ST437, showed that the A28006 strain elicited produc-tion of high levels of IL-1β, IL-12 and TNF-α, and of pyroptotic cell death in murine macrophages while the A54970 strain promoted release of high levels of IL-10 and low levels of IL-1β production, also by mouse macrophages [13]. Also, macrophages which had contact with the A54970 strain and were later treated with K. pneumoniae LPS were still unable to produce IL-1β. Moreover, strain A54970 was also able to inhibit inflammasome activation due to its ability to promote IL-10 secretion, resulting in bacterial survival and dissemination [13]. The different inflammatory phenotypes induced by these two CRE K. pneumoniae strains might be caused by either diverse capsular poly-saccharides or lipopolysaccharides.”.

2 – Material and methods: Please give a brief description of NGS in section 2.2. Please describe briefly the cell viability assay used in the experiments in section 2.5. Please describe briefly the isolation of the monocytes in section 2.6. Please give the exact serotype of E. coli LPS used as controls.

We thank the reviewer for his/her suggestions and we have now modified the text as suggested.

3 – The authors described the translocation of the p65 protein of the NFkB pathway. The translocation of p50 would be also informative since p50 in homodimers ia also able to activate the transcription of the pro-inflammatory cytokines. Did the autohors consider that other signalling pathways e.g. C/EBPβ may contribute to cytokine production?

We thank the reviewer for his/her comment. As replied to reviewer 1 and, as suggested by the same, the NF-κB data was moved to supplementary data, due to its preliminary nature. However, we agree with your comments and have now included the following in the Discussion: “ However, it has been shown that NF-κB shuttles back and forth from the nucleus to the cytoplasm following stimulation with LPS [45]. The translocation of NF-κB monomers other than p65, such as p50, and their consequent dimerization, may also activate the transcription of pro-inflammatory molecules [46]. In addition, p65 is a crucial factor of the NF-κB ATF3-CEBPδ transcriptional pathway, which modulates the transition be-tween transient to persistent and vice-versa of TLR4 stimulation in macrophages [46]. To fully understand the effect of K. pneumoniae LPS O2afg-antigen in modulating im-mune responses through NF-κB, these factors must also considered and future studies will unravel their role in this immune evasion mechanism.”.

4 – On which basis did the authors used 1 ug/mL of LPS at the treatments? Did the authors carry out concentration dependence experiments?

We understand the reviewer’s concern. Yes, we performed set-up experiments using different LPS concentrations and time-points. We started with an LPS stimulation using 0,01, 0,1, 1,0 and 10 ug/ml for 6, 12, 24 and 48 h, according to the available literature. The treatment with 1ug/ml was the best concentration to use for cytokines’ release based on the time points that we have chosen, allowing us to study the effects of LPS on the highest number of cytokines, since production of each cytokine can peak at different time-points.

5 – How the authors explain that the IL-1β level increased in case of O2afg, meanwhile the NFkB translocation decreased?

We thank the reviewer for this pertinent comment. It has been published that inhibition of IKKβ (and consequent inhibition of NF-κB) can also increase IL-1β secretion following LPS treatment (Greten F et al, Cell, 2007), and we have now added in the Discussion the following: “Moreover, studies using mouse macrophages have shown that inhibition of IKK-β (and consequent inhibition of NF-κB) can also increase IL-1β secretion following LPS treat-ment [41], which can explain our results, where also an increase of IL-1β levels was observed following treatment with LPS O2a and LPS O2afg (Fig. 3E).”.

6 – The IL-1β level increased in case of O2afg as well as the phagocytic activity of monocytes. Is it possible to have connection between them? Please discuss this issue in the manucript.

We thank the reviewer for his/her suggestion and we have now included in the Discussion the following about this topic: “An intriguingly result is the ability of LPS O2afg to induce a high production of IL-1β, which further increases after incubation with PMB, a molecule which inhibits TLR4 pathway [42]. Moreover, we showed that the 6613 strain is eagerly phagocyted (Fig. 4A-C). Combined with other factors such as capsular polysaccharides, the ability of LPS O2afg of 6613 strain to induce IL-1β secretion may initiate the differentiation of monocytes into macrophages that express phagocytic markers such as CD16, CD36, CD64 and CD206, through the TLR2/1 pathway [44], resulting in the high levels of phagocytosis observed for this particular K. pneumoniae strain.”.

7 – Is it possible that the different O antigens act on the inflammasome? Please discuss this in the manucript.

We agree with the reviewer and have now discussed it. Please see reply to your comment #1.

8 – Point and comma are equally used for the decimal signs e.g lines 134; 217; 219; 232, versus lines 115, 118, 119; 125.

We thank the reviewer for having noticed that and all decimal signs have been changed to a comma.

9 – Figures (Figure 1-3) are needed to be reconsidered, since the font size is very small, and maybe only in the pdf version, but they are in vertical form. Please re-organize the structure of the figures because the current form is not friendly for readers.

We agree with the reviewer and have now re-arranged the figures to make them more friendly for the readers.

Round 2

Reviewer 3 Report

All issues were answered by the authors.

Author Response

We thank the reviewer for his/her positive evaluation.